

# The relationships between exercise and affective states: a naturalistic, longitudinal study of recreational runners

Tim Bonham,  Gillian V. Pepper and  Daniel Nettle

Centre for Behaviour and Evolution, Newcastle University, Newcastle, United Kingdom

## ABSTRACT

**Background**.  Although people generally feel more positive and more energetic in the aftermath of exercise than before, longitudinal research on how exercise relates to within-person fluctuations in affect over the course of everyday life is still relatively limited. One constraint on doing such research is the need to provide participants with accelerometers to objectively record their exercise, and pagers to capture affective reports.

**Aims**. We aimed to develop a methodology for studying affect and exercise using only technology that participants already possess, namely GPS running watches and smartphones. Using this methodology, we aimed to characterize within-individual fluctuations in affective valence and arousal in relation to bouts of exercise, and explore possible moderators of these fluctuations.

**Methods**. We recruited a sample of 38 recreational runners. Participants provided daily affective reports for six weeks using their smartphones. Information on their runs was harvested from their own GPS devices via an online platform for athletes.

**Results**. Average valence and arousal were higher on days when the person had run than on the next day, and higher the day after a run than on the days after that. Over the course of the day of a run, valence and arousal declined significantly as the time since the run increased. Physically fitter participants had more positive valence overall, and this was particularly true when they had not run recently. There was some evidence of higher-dose (i.e., longer and faster) runs being associated with lower arousal on the next and subsequent days. Gender did not moderate associations between running and valence or arousal.

**Discussion**. Our study demonstrated the potential for studying the associations between affect and exercise in a way that is precise, undemanding for participants, and convenient for researchers, using technologies that participants already own and use.

Subjects Psychiatry and Psychology, Public Health
Keywords  Affect, Mood, Running, Experience sampling, Exercise, Strava

# INTRODUCTION

It is generally acknowledged that physical activity or exercise produces an affective or feeling response, typically involving more positive affective valence, and higher levels of

Corresponding author
Daniel Nettle, daniel.nettle@ncl.ac.uk

arousal, in the aftermath of exercise than when no exercise has been completed. Much of the early literature was based on experimentally assigned exercise sessions or programmes (see *Byrne & Byrne, 1993*; *Reed & Buck, 2009* for reviews). However, concerns have been raised about the extent to which experimental findings translate into everyday life (*Gauvin, Rejeski & Norris, 1996*; *Giacobbi, Hausenblas & Frye, 2005*; *Liao, Shonkoff & Dunton, 2015*). Experimental treatments may produce demand characteristics in participant responses. Moreover, experimental participants do not exert spontaneous choice about type, timing, duration or intensity of their activity. Thus, interest has been growing in the examining the within-person relationships between physical activity and affective state in the course of spontaneous voluntary behaviour.

Two recent systematic reviews confirm that in studies of spontaneous voluntary behaviour, physical activity is associated with more positive and more energetic affective states (*Kanning, Ebner-Priemer & Schlicht, 2013*; *Liao, Shonkoff & Dunton, 2015*). Kanning and colleagues (*2013*) stress three key methodological prerequisites for informative work on this topic. First, the measurement of physical activity should be objective, since self-report has proven unreliable (*Prince et al., 2008*). Second, the measurements of affective states should be instantaneous rather than, for example, based on retrospective recall at the end of the day or week. Third, the method of collecting the affective reports needs to objectively record the time at which the response was actually made, to avoid, for example, participants "back-filling" paper-and-pencil responses just before the end of the study.

Only seven of 22 papers reviewed by *Kanning, Ebner-Priemer & Schlicht (2013)* met all three of these criteria. Likewise, only six of the 14 papers included in the more recent review by *Liao, Shonkoff & Dunton (2015)* both assessed physical activity objectively and captured momentary affective reports electronically with a precise time-stamp. Moreover, in those six studies, the monitoring period was often very short (as little as one day). Thus, the ability of these studies to examine variability in, and moderators of, the affective response to exercise, is limited.

Many of the previous studies that have met *Kanning, Ebner-Priemer & Schlicht*'s (*2013*) criteria have done so by providing participants with accelerometers for measuring physical activity, and pagers or other devices for providing momentary affective assessments when prompted. This is expensive for researchers. However, many people already possess smartphones that can be used to prompt them to provide momentary affective reports. Moreover, their smartphones, or alternatively, GPS-enabled running watches, can already capture the data required to objectively characterise certain kinds of physical activity. It is popular amongst recreational runners and cyclists to record data on their activity (time, location, distance and speed), and to upload it to online platforms such as Strava (http://www.strava.com), which describes itself as 'the social network for athletes'. Many available devices can be set to upload data to such platforms automatically. Thus, in the present study, we explored the potential for using participants' own technology for objective recording of their exercise bouts, and momentary electronic assessment of affective states, in a longitudinal study of recreational runners.

As well as establishing the feasibility of using participants' own technology to capture data, we wished to explore possible moderators of the affective response to exercise.

The response may or may not be moderated by gender (*Roth, 1989*; *Rocheleau et al., 2004*), and may or may not be moderated by the individual's level of physical fitness (*Steptoe & Bolton, 1988*; *Ekkekakis & Petruzzello, 1999*; *Blanchard et al., 2001*). All possible forms of moderation of the affective response by exercise dose (that is, exercise intensity and duration) are represented in the literature: higher doses produce more positive effects (*Rocheleau et al., 2004*); lower doses produce more positive effects (*Ekkekakis & Petruzzello, 1999*); or intermediate doses produce the most positive effects (*Berger & Motl, 2000*). This may be partly an issue to do with elapsed time. The affective response to exercise attenuates over the scale of several hours (*Wichers et al., 2012*), but the attenuation rate may be different for different doses (*Bixby, Spalding & Hatfield, 2001*). To address the issue of the relationship between exercise dose, time, and affect, it is necessary to have quite long time series from each participant, to provide within-subject variation in exercise dose, and also in the time delay between exercise bout and affective report.

With these objectives in mind, we completed a six-week study of an opportunity sample of recreational runners who used the Strava platform. We polled affective states via a daily smartphone message, on a predetermined schedule unrelated to the schedule of exercise. Affective states were characterized based on the circumplex model of affect (*Rusell, 1980*; *Posner, Russell & Peterson, 2015*), which places momentary states on the two independent continua of valence (negative to positive), and arousal (deactivated to activated or energetic). Data on exercise bouts were separately harvested from the participants' profile on http://www.strava.com. This gave use precise and objective information on exercise bouts without placing any additional demands on participants.

Our study questions were the following. First, what is the relationship between current affect (valence and arousal) and the time since the last run? Given previous findings about the affective response to acute exercise, we would expect valence and arousal to be highest on average in the immediate aftermath of a run, and to reduce progressively as time passes. Second, do dose of running, level of physical fitness, or gender moderate the relationships between affective state and time? For example, the affective response to a high-dose run may have a different time course than the response to a low-dose run, and the magnitude and timing of affective responses to running may be different in fitter versus less fit individuals.

## MATERIALS & METHODS

### Ethics information
All participants provided electronic confirmation of their informed consent for all aspects of participation, including active permission for the researchers to follow their runs via the Strava platform. Data were managed, documented, and stored according to the Newcastle University Records Management policy. Ethical approval was granted by the Faculty of Medical Sciences research ethics committee, Newcastle University, reference number 2805/2015.

### Participants and recruitment
We used personal networks and social media to recruit an opportunity sample of recreational runners in Northern England. Potential participants were eligible if they

(1) were between 18 and 50 years old; (2) used the Strava platform to record their runs; and (3) had registered a parkrun time within the twelve months prior to study registration. Parkrun is an organization providing free, weekly timed 5 km runs in parks across Britain. We used the participant's parkrun time, relative to age- and gender-specific norms, to establish their level of physical fitness. Necessitating previous completion of a parkrun event meant the study population did not include any entirely novice runners, although parkrun represents a large cross-section of the population, attracting for example a substantial representation of women, older adults and overweight individuals (*Stevinon & Hickson, 2014*). An incentive of entry into a prize draw for £100 in running-shop vouchers was offered to participants. We recruited 40 participants. Two participants did not complete any runs during the six weeks, and henceforth have been excluded.

## Measurement of valence and arousal

Current affect was measured daily by random selection of one of four polling times: 09:00, 13:00, 17:00, or 21:00. At the selected time, an SMS message was delivered to the participant's smartphone containing a link to a very brief questionnaire to be completed on the smartphone via a web browser. The time of the affect information was taken as the time the response was received.

At each sampling point, we measured both valence and arousal (*Kragel & LaBar, 2016*). Valence describes the subjective positive-negative evaluation of a feeling state, whilst arousal is a subjective evaluation of how stimulated one feels. We used the Feeling Scale (*Hardy & Rejeski, 1989*) to measure valence. This consists of a single item of an 11-point bipolar scale ranging from −5 to +5: 'Very bad' to 'Very good'. Arousal was measured with Felt Arousal Scale (*Svebak & Murgatroyd, 1985*), a single item response on a 6-point scale anchored with 'Low arousal' and 'High arousal', prefaced by an explanation of what arousal means. The differences in structure of the two scales decreases the chances of their results being artificially correlated and improves discriminant validity by enforcing users to consider both questions separately (*Ekkekakis, 2013*).

In common with other studies relating exercise and affect (*Hall, Ekkekakis & Petruzzello, 2002*; *Parfitt, Rose & Burgess, 2006*; *Rose & Parfitt, 2007*), our measures consisted of a single item. Although multiple-item measures are less susceptible to sources of random measurement error, they are time consuming to complete, and thus increase the risk of non-compliance, particularly in the current case where each participant provided many responses. The Feeling Scale and Felt Arousal Scale were designed to be concise enough to be practical for use in the study of acute effects of exercise. The level of compliance was high: valence responses were received for 1,331 of a possible 1,596 participant-days, and arousal responses for 1,315. This amounts to means of 35.03 (s.d. 7.77) and 34.61 (s.d. 7.62) reports per participant respectively, from a possible 42.

## Running data

The Strava platform allowed us to see when a participant had completed a run, as well as the run's GPS-measured distance, pace, and Grade Adjusted Pace (GAP). GAP is pace adjusted for variations in gradient, so that hilly runs are accorded the GAP that the same

work would have achieved on the flat. In instances where a participant completed two runs on the same day (32 occasions involving 10 participants), only the run completed in closest proximity to the next affective report was counted in the analysis. The most recent run was thus presumed to have the biggest impact on subsequent affect. To provide a single number of the dose of the run, we calculated the distance (km) divided by the GAP (mins). Dose as we define it here (the product of speed and distance) provides a good summary of the strenuousness and physiological cost of exercise (*Kesaniemi et al., 2001*).

### Time since run

We divided all affective reports into *run day* (i.e., received on a day the participant had already run); *next day* (on the day after completion of the most recent run); and *baseline* (two or more days since the last run). For *run day* reports, we also used the time between the end of the run and the receipt of the affective report, in minutes.

### Moderating variables

Gender was reported at initial recruitment. *Running fitness* was assessed from the participant's best parkrun time in the past 12 months, using the parkrun organization database's (http://www.parkrun.org.uk) age grade feature. The age grade compares participant's parkrun 5 km time to the age-group and sex-specific world records for the 5 km (the majority of participants, both male and female, were in the 20–24 age category; three men and three women were over 40, and none was older than 49). Thus, a running fitness score of 100% equates to the world-record 5 km performance for a person of that age and sex; a score of 50% means they took twice as long as the world record. A person's 5 km time trial time is highly correlated with $VO_2$ max, recognised as the ultimate measure of cardio-respiratory fitness (*Ramsbottom et al., 1989*).

### Data analysis

Data were analysed in R (*R Core Development Team, 2015*). The raw data and R script for performing the analysis are publically available via the Zenodo repository at: https://doi.org/10.5281/zenodo.847176. The main analyses used linear mixed models where the outcome variable was affect (either valence or arousal, as specified in Results), with a random effect of participant to account for the repeated measures from the same participants. Around 15% of the variation in Valence, and around 20% of the variation in Arousal was between subjects, with the remainder within subjects (as estimated from a linear mixed model with no fixed predictors).

In our first model for each outcome variable (model 1), we entered day type (i.e., run day/next day/baseline) as the sole fixed predictor. To investigate whether running caused changes in affect, rather than for example participants going for a run on days when they were already feeling positive, we then restricted analysis to days when one run occurred, and compared affective outcomes if the report was received after the run rather than before it (model 2). Our third model (model 3) investigated possible moderators of the relationship between day type and affect. We started with a maximal model containing day type, running fitness, dose, gender, and the two-way interactions between these additional variables and day type (i.e., day type*running fitness, day type*dose, day type*gender).

 

**Table 1  Descriptive statistics for the main study variables.** Means, standard deviations and ranges, or numbers of participants, are shown. Note that standard deviations for valence, arousal and dose reflect the variation between reports, not between participants.

| Variable | Descriptive statistics |
| --- | --- |
| Gender | Male 20, Female 18 |
| Age groups | 18–24 years: 26; 25–34 years: 3; 35–49 years: 9 |
| Best 5k time | 64.07% (s.d. 8.05%, range 45.18–84.33) |
| Valence | 2.22 (s.d. 2.17, range −5–5) |
| Arousal | 3.28 (s.d. 1.33, range 1–6) |
| Dose | 1.73 (s.d. 1.08, range 0.18–9.3) |
| Day type | Baseline 831, Next day 457, Run day 315 |

As this model was very complex, we used stepwise model selection based on the AIC (an index of model fit) to reduce it to the optimal model. This was implemented using the R function stepAIC from the package MASS (*Venables & Ripley, 2002*). Running fitness and dose were standardized (i.e., rescaled to have a mean of 0 and a standard deviation of 1) for this model, to facilitate the interpretation of interaction terms.

Next, for run days only, we modelled valence and arousal with the time since the run in minutes as the sole predictor (model 4). Due to the reduced statistical power arising from restricting analysis to run days, no additional moderators were included in this analysis.

To provide estimates of the effect size of within-individual predictors in the linear mixed models, we calculated the proportional reduction in variance (PRV) due to each (*Peugh, 2010*). This is obtained by comparing the residual (i.e., report-level) variance of a model containing the predictor to that of a model excluding the predictor, hence providing an indication of the percentage of the variability in affective responses that is explained by the predictor.

## RESULTS

### Descriptive statistics

Descriptive statistics for the main study variables are provided in Table 1. Runs were recorded on 545 of 1,331 participant-days with valence data. Individual participants recorded 2–45 runs during the study period (mean 16.75, s.d. 9.03). Examining correlations at the between-subject level, there was a significant correlation between participants' running fitness and their average dose ($r = 0.52, p < 0.01$), and the correlation between running fitness and the total number of runs in the study period was marginally non-significant ($r = 0.31, p = 0.06$). Total number of runs was not significantly correlated with average dose ($r = 0.13, p = 0.45$).

Valence and arousal were not strongly related to one another. There was evidence of an inverted-V shaped relationship between the two on average, as has been observed elsewhere (*Kuppens et al., 2013*). That is, average arousal was highest at the negative and positive extremes of valence, and lowest at neutral valence (Fig. 1). However, the association between valence and arousal was fairly weak, with the absolute value of valence accounting

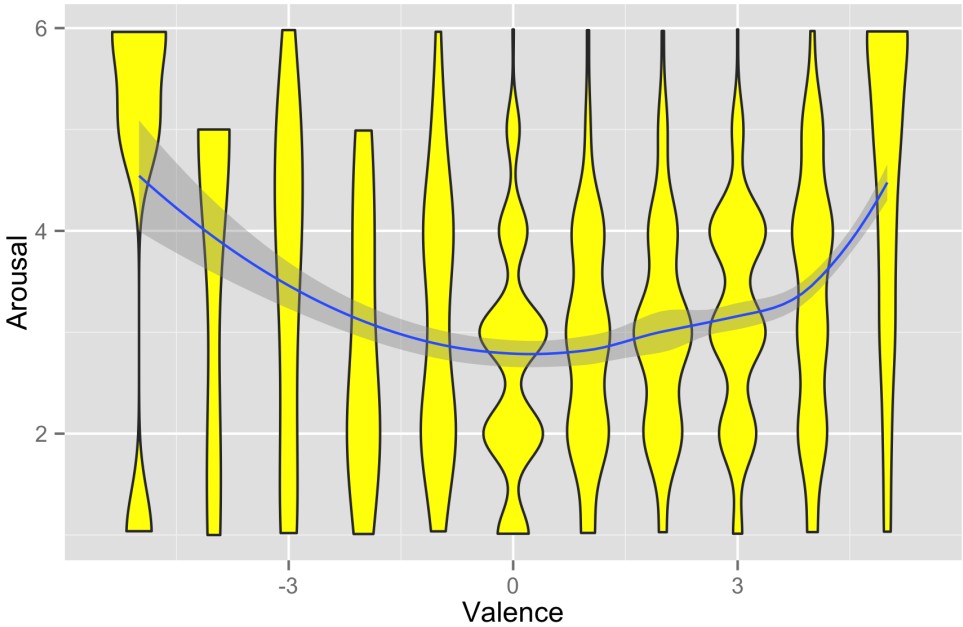

**Figure 1** **Relationship between affective valence and arousal in the data set.** The polygons show the density of arousal values observed at each value of valence. The solid line and grey shading represent the mean of arousal as a function of valence (and the 95% confidence interval), calculated using locally-weighted polynomial regression.

for only 12% of the variation in arousal. For this reason, we henceforth treat valence and arousal as two independent outcomes, as is conventional in the affect literature (*Barrett & Russell, 1999*).

Average affect did not differ substantially with time of day. Defining morning as before noon, afternoon as noon to 6 pm, and evening from 6 pm, valence was not significantly different across the morning (mean 2.06, s.d. 2.25), afternoon (mean 2.20, s.d. 2.07) and evening (mean 2.36, s.d. 2.25; linear mixed model with morning as the reference category: $t_{afternoon} = 0.94$, $p = 0.35$; $t_{evening} = 1.89$, $p = 0.06$). Likewise, arousal was not significantly different across the morning (mean 3.38, s.d. 1.39), afternoon (mean 3.31, s.d. 1.28) and evening (mean 3.18, s.d. 1.36; linear mixed model with morning as the reference category: $t_{afternoon} = -0.88$, $p = 0.38$; $t_{evening} = -1.80$, $p = 0.07$). Thus, we felt justified in not including time of day of report in subsequent analyses.

## Valence in relation to running
Our first model predicted valence by day type. Mean valence was significantly higher on run days than next days, and also significantly higher on next days than baseline days (Table 2, model 1; PRV due to time since run 6.9%). This is illustrated in Fig. 2A. The higher valence of run days could reflect either direction of causation: running increases valence, or more positive valence makes going for a run more likely. To distinguish these two possibilities, we compared valence on the day when a run occurred and the report

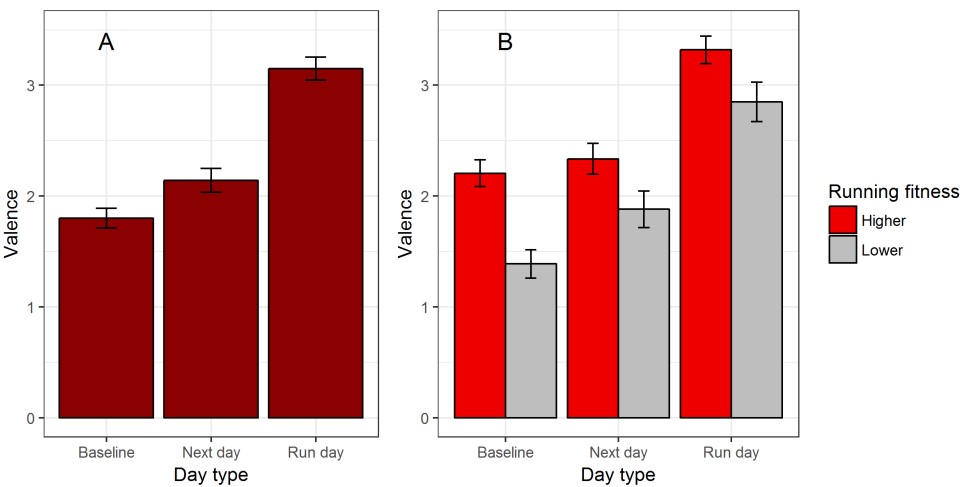

**Figure 2   Valence in relation to running.** (A) Mean valence by day type. (B) Mean valence by day type, split by whether the participant's running fitness (age- and gender-adjusted best 5 k time) is above or below the median for the sample. Error bars represent the standard error of affective reports.

**Table 2   Summary of statistical models with valence as the outcome variable.** All models contain a random intercept for participant. Model 2 is restricted to days where one run occurred, and model 4 to days where a report was received after a run.

| Model | Predictor | B | 95% CI | | t | p-value |
|---|---|---|---|---|---|---|
| 1 | Day type: Baseline | Reference category | | | | |
| | Day type: Next day | 0.26 | 0.01–0.51 | | 2.00 | 0.046* |
| | Day type: Run day | 1.38 | 1.10–1.66 | | 9.65 | <0.01* |
| 2 | Before run | Reference category | | | | |
| | After run | 1.21 | 0.89–1.52 | | 7.49 | <0.01* |
| 3 | Day type: Baseline | Reference category | | | | |
| | Day type: Next day | 0.26 | −0.23–0.76 | | 1.03 | 0.305 |
| | Day type: Run day | 1.16 | 0.80–1.53 | | 6.21 | <0.001* |
| | Dose | 0.01 | −0.35–0.38 | | 0.07 | 0.940 |
| | Running fitness | 0.65 | 0.22–1.07 | | 3.05 | 0.004* |
| | Next day*Dose | −0.50 | −1.10–0.11 | | −1.59 | 0.111 |
| | Run day*Dose | 0.21 | −0.19–0.62 | | 1.02 | 0.309 |
| | Next day*Fitness | −0.77 | −1.37–−0.18 | | −2.52 | 0.012* |
| | Run day*Fitness | −0.62 | −1.02–−0.22 | | −3.03 | 0.003* |
| 4 | Time since run (minutes) | −0.001 | −0.002–−0.001 | | −2.98 | 0.003* |

**Notes.**
*$p < 0.05$.

was before the run, and when the report was after the run (Table 2, model 2). Valence was significantly higher after than before the run (after: mean 3.19, s.d. 1.77; before: mean 2.08, s.d. 2.04; PRV 10.1%). This supports running being causal in increased valence.

Next, we examined possible moderators of the relationship between valence and day type using the stepwise model selection procedure described in Methods. Gender and its

interactions were eliminated in the model selection, and so the final model contained day type, running fitness, dose, and the interactions of best 5 k time and dose with day type (Table 2, model 3). The magnitudes of the parameter estimates for the differences between run days, next days and baseline was similar to those in model 1. Although the interaction of dose and day type was retained by the model selection procedure, neither of the parameter estimates describing this interaction differed significantly from zero (Table 2, model 3). There was a significant main effect of running fitness, with higher running fitness time associated with higher average valence (Table 2, model 3). There was also significant interaction between day type and running fitness (Table 2, model 3, PRV 1.8%): the magnitude of the difference between participants high and low running fitness was larger on baseline days than on other days (Fig. 2B).

For run days only, we modelled valence in relation to time in minutes since the completion of the run. Time since run was a significant predictor (Table 2, model 4; PRV 2.7%). The parameter estimate corresponds to a reduction in valence of 0.07 units on the scale (or 0.04 standard deviations) with each hour that passes after the run (Fig. 3). This estimate was not substantially altered by additional controlling for time of day.

## Arousal in relation to running

Our first model predicted arousal from time since run alone. Arousal was significantly higher on run days than baseline, but baseline days and next days did not differ significantly (Table 3, model 1; Fig. 4A). This is consistent with running producing a short-term increase in arousal, though the effect size (PRV 0.6%) was much smaller than for valence. We once again tested the direction of causation by comparing arousal on run-days where the report came after the run as opposed to before (Table 3, model 2). There was not a clear significant difference (after: mean 3.54, s.d. 1.39; before: mean 3.48, s.d. 1.26; PRV 0.2%). Thus, it is possible that participants were motivated to go for a run when already feeling aroused, or became more aroused in anticipation of a planned run, rather than, or in addition to, running increasing arousal.

We next performed the model selection procedure on possible models including dose, fitness, gender and their interactions with day type. Gender, running fitness and their interactions were excluded in the model selection, so the final model contained day type, dose, and their interaction (Table 3, model 3; PRV 1.6%; although the day type by dose interaction was retained by the model selection, the parameter estimate was marginally non-significantly different from zero, $p = 0.06$). The interaction between day type and dose appeared to be driven by the pattern of lower arousal on next and baseline days than on run days only being present for high-dose runs. For low-dose runs, next day and subsequent arousal appears to be no lower than on the day of the run (Fig. 4B).

For run days only, we again modelled arousal in relation to time in minutes since the completion of the run. Time since run was a significant predictor (Table 3, model 4; PRV 1.7%). The parameter estimate corresponds to a reduction in arousal of 0.04 units on the scale (or 0.03 standard deviations) with each hour that passes after the run (Fig. 5). This estimate was not substantially altered by additional controlling for time of day.

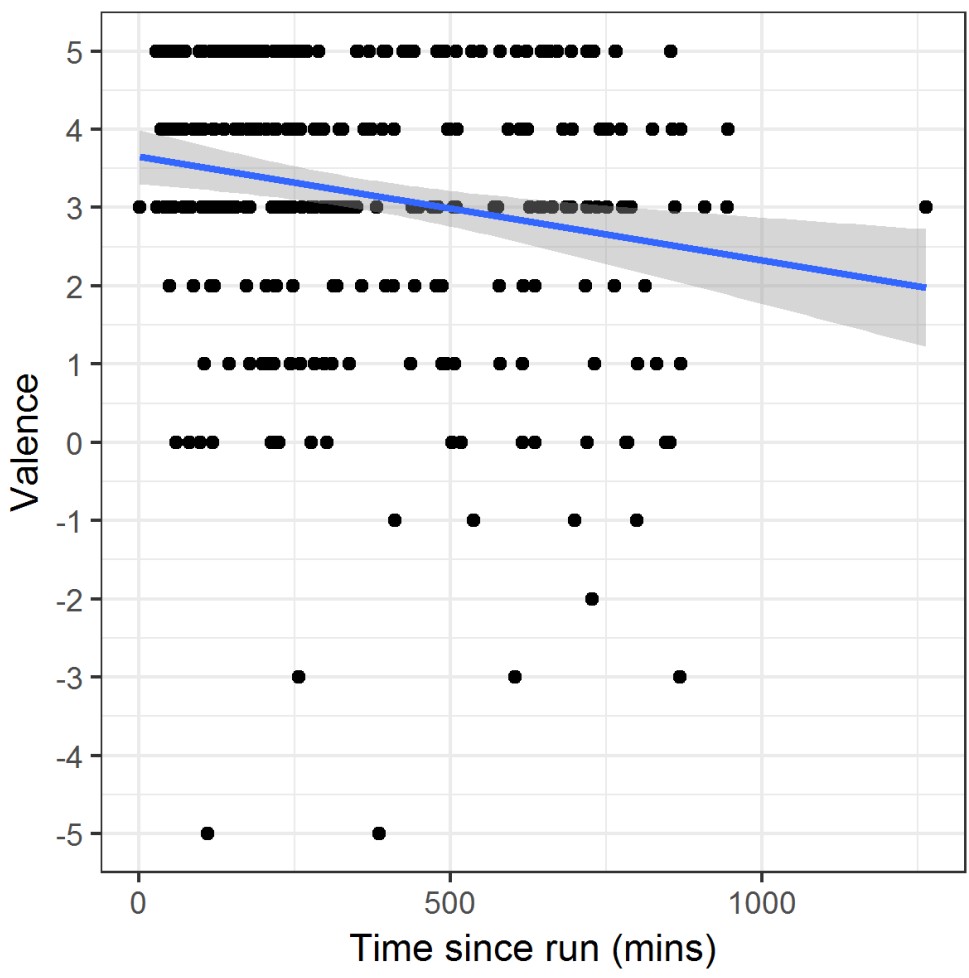

**Figure 3** **Valence against time since completion of the run on run days.** Points represent individual reports. The line and shaded region represent a linear fit and its 95% confidence interval.

## DISCUSSION

In a six-week study of how valence and arousal fluctuate in relation to running, we found clear evidence that average affective states are related to the occurrence of runs. Average valence was highest on days when the participant had been for a run, and this appeared to be due to running increasing valence rather than high valence initiating a run. The valence premium declined with each hour that passed as the day went on. Taking the parameter estimates from model 4 of valence, it would take approximately four hours for valence to decline by an amount equivalent to the difference between the average run-day valence and the average next-day valence. Thus, our findings are consistent with evidence that having done exercise has a positive acute effect on valence, an effect that attenuates over the course of hours (*Reed & Ones, 2006*; *Wichers et al., 2012*; *Liao, Shonkoff & Dunton, 2015*). We did however find that valence was still slightly higher the day after a run compared to baseline. This suggests either that the valence effects of running are not completely dissipated in

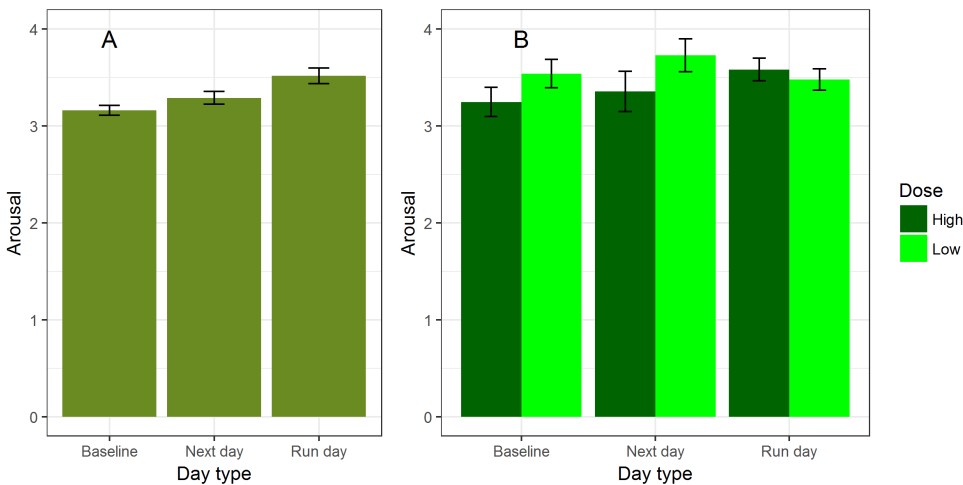

**Figure 4 Arousal in relation to running.** (A) Mean arousal by day type. (B) Mean arousal by day type, broken down by whether the most recent run was higher or lower than the median dose. Error bars represent the standard error of affective reports.

**Table 3 Summary of statistical models with arousal as the outcome variable.** All models contain a random intercept for participant. Model 2 is restricted to days where one run occurred, and model 4 to days where a report was received after a run.

| Model | Predictor | B | 95% CI | t | p-value |
|---|---|---|---|---|---|
| 1 | Time: Baseline | Reference category | | | |
| | Time: Next day | 0.01 | −0.15–0.16 | 0.06 | 0.949 |
| | Time: Run day | 0.24 | 0.07–0.42 | 2.74 | 0.006* |
| 2 | Before run | Reference category | | | |
| | After run | 0.09 | −0.12–0.30 | 0.88 | 0.380 |
| 3 | Time: Baseline | Reference category | | | |
| | Time: Next day | −0.03 | −0.35–0.30 | −0.17 | 0.864 |
| | Time: Run day | 0.12 | −0.13–0.36 | 0.94 | 0.347 |
| | Dose | −0.15 | −0.39–0.09 | −1.24 | 0.217 |
| | Next day* Dose | −0.14 | −0.54–0.26 | −0.69 | 0.499 |
| | Run day* Dose | 0.26 | −0.01–0.52 | 1.88 | 0.061 |
| 4 | Time since run | −0.001 | −0.001–−0.000 | −2.28 | 0.023* |

**Notes.**
*$p < 0.05$.

24 h, or else that, for these habitual runners, not running for more than a day or two tends to be caused by other factors that could cause low valence, such as illness or work demands.

Higher-fitness participants (as measured by their best performance for age and gender in a 5 km timed run), had higher valence overall than low-fitness participants. One simple explanation for the positive effects of long-term physical exercise participation on affect is just the repeated reinstatement of acute effects by exercise bouts (the "maintenance" explanation; *Reed & Buck, 2009*). This explanation would predict that fitter runners feel more positive at any given time simply because they are more likely to have run recently.

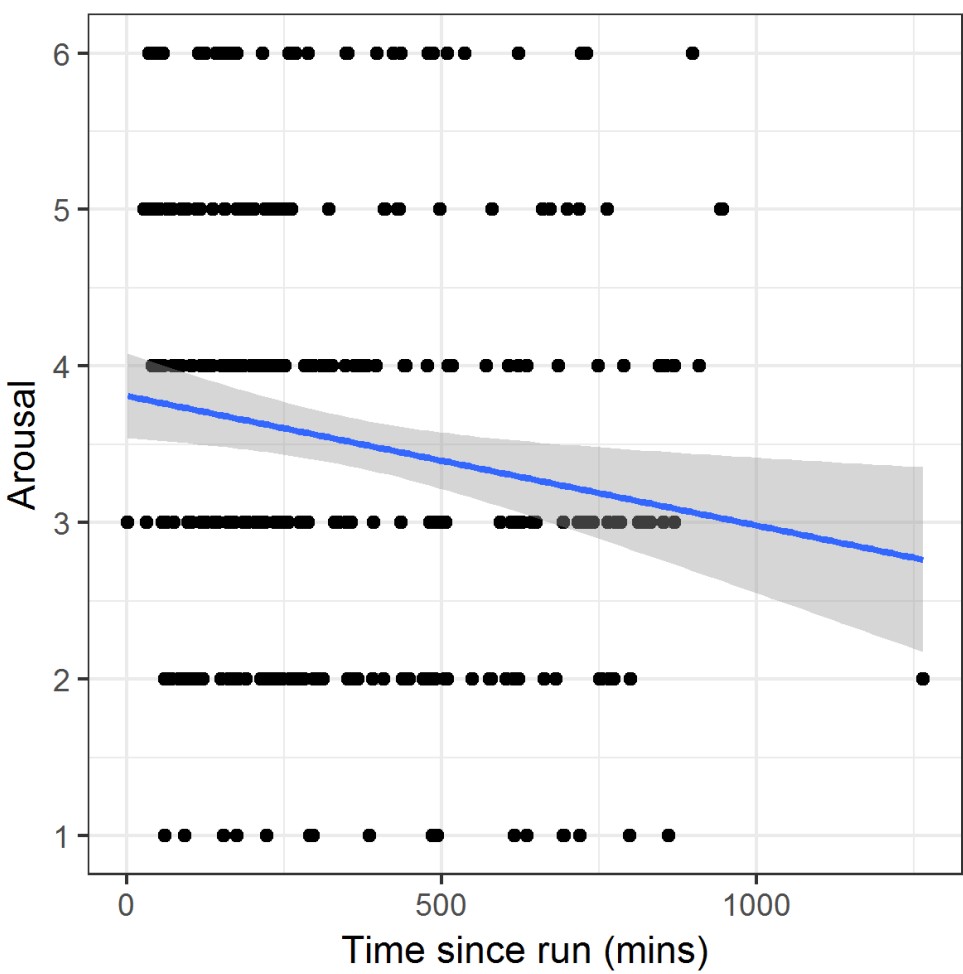

**Figure 5 Arousal against time since completion of the run on run days.** Points represent individual reports. The line and shaded region represent a linear fit and its 95% confidence interval.

However, this cannot be the complete explanation in our data. Fitness was associated with more positive valence even after controlling for day type. Moreover, the biggest valence difference between fitter and less-fit individuals was on baseline days; that is, when they *hadn't* run recently. This suggests that there may be long-term affective adaptation that comes with physical fitness, as a result of training or associated lifestyle factors. For example, fitter individuals have reduced chronic pain, reduced body weight and improved immune function (*Nielens & Plaghki, 2001*; *Warburton, Nicol & Bredin, 2006*), all of which could contribute to more positive valence at baseline.

Arousal was also higher in the immediate post-run period than at any other time. However, the association between arousal and time since run was weaker than that for valence, with time since run explaining less than 1% of report-to-report variation in arousal. Moreover, arousal was not clearly higher after compared to before the run on days when a run occurred. This suggests that the association between arousal and the timing of runs could be at least partly due to people choosing to run when their arousal is high, or

their arousal rising in anticipation of a forthcoming run, rather than exclusively due to the run itself increasing arousal. Arousal did however reduce with increasing time since the run on run days, at a rate suggesting it would take approximately 6 h for the arousal premium of a run day to dissipate. We also found that running dose moderated the relationship between arousal and time since run. Specifically, a higher dose of running (compared to a lower dose) was associated with similar arousal on the day of the run, but lower arousal the next day and on subsequent days. Unfortunately it is impossible to distinguish with our data whether this is a general effect of high-dose exercise, or something specific about the particular high doses that these runners completed (for example, that the highest doses in the dataset represented races to which the runners had been building up, after which there was a drop in arousal, rather than regular training runs).

Although our main results support the widespread consensus in the literature that physical exercise can lead to higher affective valence and increased arousal, our findings differ in some regards from previous findings in the literature. First, we found no evidence that—in this sample—higher doses of running are associated with a less positive (or even negative) effect on affect, compared to lower doses (*Ekkekakis & Petruzzello, 1999*; *Reed & Ones, 2006*). The only negative effects of dose for which we found any evidence were possibly in the days after the run, on arousal. This may reflect the difference between interventional studies that assign individuals to exercise, and more naturalistic studies such as ours where the participants self-selected their running dose on each occasion. It is also important to recall that all our reports were gathered after the exercise had finished. Measurements collected during exercise might have produced a different picture (*Steptoe & Bolton, 1988*). Thus, our findings are not inconsistent with the idea that if individuals were assigned very high doses, this might produce a negative affect response, or that they might feel bad during the effort; they just show that in the course of spontaneous self-directed running, valence and arousal are more positive on average after a run than at any other time, regardless of dose. Our data did not support the possibility that the affective response to running would be different for men and women (*Rocheleau et al., 2004*).

It is important to acknowledge both the strengths and limitations of our design. As our study was observational and only included habitual runners, it is not suitable for estimating the affective impact that would follow if current non-runners began to run. This is because the population sub-group that runs habitually is likely to be biased towards those who have a more positive affective response to running (*Hoffman & Hoffman, 2008*; *Williams et al., 2008*; *Hallgren, Moss & Gastin, 2010*). Whilst the non-experimental approach limits the causal inferences that can be made, it has high ecological validity. Experimental studies may assign doses of exercise that individuals would not choose at times they do not wish to do them, and there is always the risk of experimental demand characteristics. Our dataset thus allows more naturalistic insight into the role that running plays in fluctuations of affective state over time in the everyday lives of recreational runners.

The combination of gathering affect data via experience sampling, and using a social network for athletes to automatically record running using their own GPS devices, was extremely effective. Participant demand was light, since our affective measures were very quick to complete, and running data would have been uploaded to Strava anyway. Thus,

participant compliance was good, and the running data had greater precision than would be feasible with manual reporting. It also made it unnecessary for the researchers to interfere in any way in participants' self-selected running schedules. However, the methodology did produce some specific limitations. The schedule of reporting was completely independent of the schedule of running. Thus, we relied on reports happening to fall into various time windows after runs. This means that, unlike *Gauvin, Rejeski & Norris (1996)*, we could not make explicit immediately-before/immediately-after affect comparisons. Participants were not prevented from engaging in other forms of exercise during the study; these would not have been captured, and nor would any run where the participant had for some reason not used their device. Thus, by failing to account for the fact that certain affective reports may have come in the aftermath of non-recorded bouts of physical activity, we may if anything have underestimated the affective response to activity.

Moreover, although the running data was precise, it was not very rich. Some runs may have been completed socially, but Strava data did not allow us to identify these, and hence test whether social running has different affective consequences than non-social running (*Cruwys et al., 2013*). Similarly, we made no attempt to distinguish runs taking place in different environments, such as on green space versus roads.

## CONCLUSIONS

Our naturalistic study suggested that, in recreational runners, running provides a boost to both valence and arousal, a boost that is larger for valence than for arousal, and is of around the same magnitude for men and women. The boost dissipates over the course of hours, and high doses may produce a relative slump in arousal on the next day and subsequent days compared to low doses. We also found evidence that fitter individuals have more positive valence, especially when they have not run recently. This suggests that the effects of exercise may involve not just a series of immediate valence boosts, but adaptation in baseline valence as well. Our research demonstrates the feasibility of studying affect and exercise in a way that is economical, precise, easy for participants, and convenient for researchers, using technologies that participants already own and use.

## ACKNOWLEDGEMENTS

We thank all our participants, and the Graduate School of the Faculty of Medical Sciences, Newcastle University.

### Funding

The authors received no funding for this work.

### Competing Interests

The authors declare there are no competing interests.

## Author Contributions

- Tim Bonham conceived and designed the experiments, performed the experiments, analyzed the data, wrote the paper, prepared figures and/or tables, reviewed drafts of the paper.
- Gillian V. Pepper conceived and designed the experiments, wrote the paper, reviewed drafts of the paper.
- Daniel Nettle conceived and designed the experiments, analyzed the data, wrote the paper, prepared figures and/or tables, reviewed drafts of the paper.

## Human Ethics

The following information was supplied relating to ethical approvals (i.e., approving body and any reference numbers):

Ethical approval was granted by the Faculty of Medical Sciences research ethics committee, Newcastle University, reference number 2805/2015.

## Data Availability

Bonham, Tim, Pepper, Gillian, & Nettle, Daniel. (2017). Data and script for 'The relationships between exercise and affective states: A naturalistic, longitudinal study of recreational runners' [Data set]. Zenodo. http://doi.org/10.5281/zenodo.847177.

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
