# Peer review of "The relationships between exercise and affective states: a naturalistic, longitudinal study of recreational runners"

_PeerJ, doi:10.7717/peerj.4257_

## Round 0.1 · original submission · Minor Revisions

Please take on board as many suggestions as possible as these will improve the impact of your study

Reviewer 1 ·

Basic reporting

The manuscript “The relationships between exercise and affective states: A naturalistic, longitudinal study of recreational runners is clear written, with relevant figures and tables, which are of high quality and well labelled and described. The structure conforms to PeerJ standards except the abstract because headings are missing.
Nevertheless, there are some issues, which should be improved. My most important issue is with the theoretical background in the section introduction and some missing information about statistical analysis.
Introduction:

Experimental design

The introduction needs more detail (references and theoretical assumptions/backgrounds) and I would restructure it in putting some information in the method section. If you add these backgrounds, you could improve your argumentation and strengthen the relevance of your research question.

Validity of the findings

data is robust and conclusion are well stated

Additional comments

In this section, I listed some comments for each section (if necessary.
Introduction:
Line 29: These references are very old, try to find current evidence of the association between exercise and affective states.
Line 53: Your intention is to analyse possible interactions. You mentioned this issue in line 53, for instance. Here, the argumentation would become stronger if you provide theoretical assumptions about these possible interactions (e.g. dual mode theory). Furthermore, you assume that valence and arousal reduce progressively as time passes. You also should add references for this assumption, for instance: Wichers et al. (2011). A time-lagged momentary assessment study on daily life physical activity and affect. Health Psychology
Line 55 (et seq.): You mentioned that the association between exercise and affective states should be assessed in naturalistic studies. There are a lots of studies analysing the association between physical activity and affect in daily life and you should define in more detail your scientific contribution (see for instance the reviews of: Kanning, M., Ebner-Priemer, U., & Schlicht, W. (2013). How to investigate within-subject associations between physical activity and momentary affective states in everyday life: A position statement based on a literature overview. Frontiers in Psychology Movement Science and Sport Psychology; Liao, Y., Shonkoff, E. T., & Dunton, G. (2015). The acute relationships between affect, physical feeling states, and physical activity in daily life: A review of current evidence. Frontiers in Psychology Movement Science and Sport Psychology.
Line 67 (et seq.): In my opinion you should explain in more detail the theoretical background of affective states (e.g. circumplex model, Russel, because you measure valence and energetic arousal)
Line 70 (et seq.): This detailed information about your study belong in the section “Methods”, not in the section “introduction”

Data analysis:
Line 153: You should add information about the within- and between person variance (intraclass coefficient) and how you have centred the variables. To support understanding you could add the equations of the models.

Results:
How many missings do you have in your affect measures. You measured affect four times a day over six weeks (4measures x 42days = 168 measurement points). In average, how many affect measures did you achieve per person?
Line 180 (et seq.): It is correct to abbreviate standard deviation with s.d.?
Discussion:
You should discuss in the limitation section that you assessed physical activity with self-reports. The activity data depends on what the participant enters into the platform. You are not able to control if the entry about time, distance and intensity of the run is correct.

·

Basic reporting

No comment.

Experimental design

Clear specification of inclusion and exclusion criteria (health status, motivations, etc.).
Specify the average age of the female and male samples.
Clear formulation of the main purpose of the research.

Validity of the findings

Specify the limits, strengths and perspectives of the research.

---

## Round 0.2 · accepted · Accept

All the points raised by the reviewers have been satisfactorily addressed.